# Effects of In Vitro Fermentation of Polysialic Acid and Sialic Acid on Gut Microbial Community Composition and Metabolites in Healthy Humans

**DOI:** 10.3390/foods13030481

**Published:** 2024-02-02

**Authors:** Zhongwei Yin, Li Zhu, Minjie Gao, Dan Yu, Zijian Zhang, Ling Zhu, Xiaobei Zhan

**Affiliations:** 1Key Laboratory of Carbohydrate Chemistry and Biotechnology, Ministry of Education, School of Biotechnology, Jiangnan University, Wuxi 214122, China; 7200201071@stu.jiangnan.edu.cn (Z.Y.); zhanzhuli@yahoo.com (L.Z.); jmgao@jiangnan.edu.cn (M.G.); 6210207009@stu.jiangnan.edu.cn (D.Y.); zzjyyyyyg@163.com (Z.Z.); zhuling_mimg@outlook.com (L.Z.); 2A & F Biotech. Ltd., Burnaby, BC V5A 3P6, Canada

**Keywords:** prebiotics, in vitro digestion, fecal fermentation, intestinal microorganism, short-chain fatty acid, differential metabolites

## Abstract

The influence of polysialic acid (PSA) and sialic acid (SA) on the gut microbial community composition and metabolites in healthy humans was investigated using a bionic gastrointestinal reactor. The results indicated that PSA and SA significantly changed the gut microbiota and metabolites to different degrees. PSA can increase the relative abundances of *Faecalibacterium* and *Allisonella*, whereas SA can increase those of *Bifidobacterium* and *Megamonas*. Both can significantly increase the content of short-chain fatty acids. The results of metabolome analysis showed that PSA can upregulate ergosterol peroxide and gallic acid and downregulate the harmful metabolite N-acetylputrescine. SA can upregulate 4-pyridoxic acid and lipoic acid. PSA and SA affect gut microbiota and metabolites in different ways and have positive effects on human health. These results will provide a reference for the further development of PSA- and SA-related functional foods and health products.

## 1. Introduction

The human colon contains many classes of the gut microbiota, which plays a critical part in maintaining the development and balance of the body [1]. The gut microbiota can resolve indigestible nutrients into bioactive molecules, such as neurotransmitters, vitamins, and fatty acids, which protect the body from pathogens [2]. Furthermore, the gut microbiota is susceptible to the influence of numerous factors, such as diet, living habits, and environment, and has a high degree of plasticity. Hence, modification of the gut microbiota through diet has great potential to promote human health and prevent and control diseases.

Shaping the gut microbiota with carbohydrates has become one of the important ways to promote human health [3]. Such substances are capable of changing the gut microbiota and metabolites, which results in beneficial effects on the human body [4]. Sialic acid (SA) is a monosaccharide with a nine-carbon chain; it has various biological functions, such as neurotransmission, antiviral activity, promotion of gut health and nutrient absorption, and acceleration of cognitive development of the brain [5,6]. In addition, SA can resist digestion and reach the colon for use by the gut microbiota [7]. Human milk oligosaccharides (HMOs) containing SA can promote probiotic growth and inhibit pathogenic bacteria colonization and thus have a significant effect on infant health [8,9]. At present, studies on the effects of dietary SA on the adult gut microbial ecosystem are still lacking [7]. In addition, countries such as those belonging to the European Union, Japan, Singapore, and Malaysia have approved the use of SA as a food ingredient. The SA produced by our laboratory using hydrolytic polysialic acid (PSA) meets the basic requirements of food safety and has been approved as a novel food in China.

PSA is a polymer formed by SA, and it makes a critical difference in the maintenance of nervous system development, function, and health [10,11]. PSA has widespread applications in biomedicine, bio-binding materials, and materials for drug embedding and sustained release [12,13,14]. In addition, polysaccharides of different molecular weights have varied effects on the gut microbiota [15,16]. As a polymer of SA, PSA has not been reported to regulate the gut microbiota in healthy people.

Under ideal conditions, the digestion of food and its influence on the gut microbiota can be investigated scientifically and accurately through human experiments. However, high costs, technical limitations, and ethical constraints cause difficulties in human experiments. Therefore, more in vitro gastrointestinal models have been developed to evaluate the digestive properties and prebiotic functions of foods. In this study, a self-developed bionic gastrointestinal reactor (BGR) was used, and it can simulate the whole process of food fermentation in the colon after the action of the digestive system [17]. BGRs have been used to investigate the fate of food and polysaccharides during digestion and their effects on the gut microbiota [18,19].

According to existing studies, HMOs containing SA have a positive effect on the infant gut microbiota, and dietary SA can change the adult gut microbiota community. Meanwhile, differences in the molecular weights of polysaccharides may have various effects on the gut microbiota. The difference between the effects of PSA and SA on the gut microbiota of healthy people is unknown, and no relevant studies have been reported thus far. We hypothesized that PSA and SA have different effects on the gut microbiota of healthy people. We hypothesized that (1) PSA and SA have different influences on gut microbial community composition and metabolites and (2) positive effects on the gut microbiota. To verify our prediction, we prepared PSA with a molecular weight of 260 kDa using a specific fermentation process and obtained SA via acid hydrolysis. Subsequently, a BGR was used to explore the digestive processes of PSA and SA, and their effects on the gut microbial community composition and metabolites of healthy people were explored using 16s rDNA high-throughput sequencing technology and non-targeted metabolomics technology. This study explored the different effects of PSA and SA on the gut microbiota in healthy people for the first time and conducted a comparative analysis to provide an innovative theoretical foundation and feasible solutions for the application of PSA and SA in the regulation of the gut microbiota. 

## 2. Materials and Methods 

### 2.1. Chemicals

Pepsin, α-amylase, gastric lipase, trypsin, bile salts, and vitamin K were from Sigma–Aldrich (St. Louis, MO, USA). Chlorine heme, NaCl, KCl, NaHCO_3_, HCl, MgSO_4_, CaCl_2_, CH_3_COONa, sorbitol, K_2_HPO_4_·3H_2_O, (NH_4_)_2_SO_4_, tryptone, FeCl_3_, MnCl_2_, ZnSO_4_, MgSO_4_, and oxalic acid were from China National Pharmaceutical Group Corporation (Beijing, China). The above reagents were of analytical grade.

Methanol, ammonium acetate, and formic acid were from Thermo-Fisher (Waltham, MA, USA), The above reagents were of liquid chromatography–mass spectrometry (LC-MS) analytical grade.

### 2.2. Preparation and Extraction of PSA and SA

Fermentation preparation of PSA: This study is based on the research of Zheng et al., with minor modifications [20]. PSA with a molecular weight of 260 kDa was prepared via fermentation of *Escherichia coli* K235 6E61 (CCTCC M208088) in a 7 L bioreactor. The fermentation medium consisted of (g·L^−1^) 60 sorbitol, 2.5 K_2_HPO_4_·3H_2_O, 4.94 (NH_4_)_2_SO_4_, 1.5 tryptone, 0.26 FeCl_3_, 0.11 MnCl_2_, 0.00051 ZnSO_4_, and 0.92 MgSO_4_. The culture conditions of fermentation were as follows: pH of 6.4 at 0–18 h and 7.4 at 18–36 h; mixing speeds of 250 r·min^−1^ at 0–12 h, 400 r·min^−1^ at 12–20 h, and 250 r·min^−1^ at 20–36 h; temperature of 37 °C; air flow of 1.2 vvm.

Extraction and purification of PSA: A certain amount of fermentation liquid was centrifuged at 4 °C and 9391× *g* for 10 min. Then, 95% ethanol was mixed with the supernatant at a volume ratio of 3:1 and left for 2 h at 4 °C. After centrifugation at 4 °C and 9391× *g* for 5 min, the precipitate was collected and redissolved in water. Finally, the supernatant was obtained after centrifugation at 4 °C and 13363× *g* for 5 min. In addition, purification was carried out using a procedure established in our laboratory [21].

Preparation of SA: SA was prepared using an acid hydrolysis method, in accordance with Wu et al.’s research, and slightly modified [22]. At 90 °C, the PSA solution underwent hydrolysis for 3.5 h in the presence of a 0.1 mol·L^−1^ oxalic acid solution, followed by separation and purification of the resulting hydrolysate.

After purification, the purity of each sample was above 98%. The obtained samples were non-toxic and harmless, thus meeting the basic standards of food safety, and have been approved as a novel food in China.

### 2.3. Simulated Digestion Experiments

The method for the configuration of simulated saliva was that described by Shi et al. [23]. The simulated saliva was composed of 6.2 g·L^−1^ NaCl, 2.2 g·L^−1^ KCl, 1.5 g·L^−1^ CaCl_2_·2H_2_O, and 5.5 mg·L^−1^ α-amylase. The digestive stage of the mouth was carried out in a 37 °C constant-temperature water-bath magnetic stirrer. Exactly 5 mL of simulated saliva was mixed with 5 mL of a 25 mg·mL^−1^ PSA solution, and oral digestion was finished after a 1 min reaction. The digestive stage of the gastric and small intestine occurred in the BGR, which consisted of the digestive (gastric and small intestine) and fermentation (colon) systems (Appendix A). The dynamic parameters of BGR were set as described by Li et al. and slightly modified [18]. The simulated gastric fluid, which contained 0.1 M hydrochloric acid, 8 mg·mL^−1^ pepsase, and 20 U·mL^−1^ gastric lipase, had pH 2.0. The sample flow after oral digestion was added to the gastric reactor, and the gastric digestion phase began. After 2 h of digestion, the gastric digestion phase was completed. The simulated intestinal fluid contained fresh pig bile and pancreatic fluid (4% *w*/*v*, containing amylase, protease, and lipase), electrolyte solution, and 1 M bicarbonate, and the pH was 6.5. The digestive sample from the stomach reactor was transported to the small intestine, where pancreatic and intestinal fluids were secreted, and intestinal digestion was initiated. After 4 h of intestinal digestion, the digestion liquid was freeze-dried to make lyophilized powder, which was stored at −80 °C for subsequent fermentation experiments (SA freeze-dried powder was prepared using the same digestion process). 

### 2.4. Preparation of Fecal Inoculum Solution

Fecal samples were collected from ten healthy volunteers (5 males and 5 females aged 21–28 years) who had no gastrointestinal diseases, followed a normal Chinese diet, had not taken probiotics or prebiotics for three months, and had not been treated with antibiotics. The subjects’ informed written consent was obtained for this experiment. On the day of sampling, we collected the fecal samples provided by volunteers during fasting in anaerobic tubes. The fecal dialysate was formulated as described by Aguirre et al. [24]. The fecal samples were mixed with an appropriate amount of dialysis solution, and impurities were removed with four layers of sterile gauze after homogenization. The filtrate was mixed with 30% glycerin at a ratio of 1:1 and then frozen in liquid nitrogen for subsequent in vitro fermentation experiments.

### 2.5. Fecal Fermentation in BGR

The experiment was conducted on three experimental groups: blank, SA, and PSA groups. The basic medium was formulated as described by Sun et al., with slight modifications [19]. For the supplemental medium, the composition of the blank group was exactly the same as that in the basic medium. While, in the PSA (SA) group, the amylum in the supplemental medium was replaced with PSA (SA), and the remaining components were the same as those in the blank group. The prepared medium was sterilized at 121 °C for 15 min. The fecal inoculum solution preserved in liquid nitrogen was incubated at 37 °C for 30 min. After sterilization, the BGR was injected with 180 mL of basic medium and 18 mL of fecal inoculum solution (10% inoculum, *v*/*v*) in a super-clean workbench. After inoculation, the BGR was transferred to the worktable, and the 37 °C water circulation and peristalsis devices were started. During fermentation, the pH sensor can dynamically add 0.5 mol·L^−1^ NaOH to the reactor to maintain pH stability (the pH was set to 5.8). N_2_ was injected into the BGR every 8 h to maintain an anaerobic environment. After 20 h of fermentation, the fecal fermentation liquid reached a stable stage. And the fecal fermentation liquid was starved for 2 h to exhaust the carbon source. Then, 40 mL of fermentation liquid was discharged, and 40 mL of supplementary medium was added to the BGR. At this point, the initial fermentation time (0 h) was recorded, and the experimental period commenced. During the experimental period (48 h), 40 mL of fermentation liquid was discharged every 12 h and sampled, and 40 mL of supplementary medium was added to maintain the stable working volume of the experiment. At 0, 12, 24, 36, and 48 h, 10 mL samples were collected. After liquid nitrogen freezing, the samples were subsequently stored in a −80 °C refrigerator for future experiments and sample delivery.

### 2.6. Determination of the Molecular Weight of PSA

The molecular weight change of PSA during digestion was measured via high-performance LC (Waters 1525 HPLC, Milford, MA, USA). The chromatographic parameters were as follows: Molecular weights were determined using an Ultrahydrogel^TM^ Linear 300 mm × 7.8 mm column, with a mobile phase consisting of 0.1 N sodium nitrate at a flow rate of 0.5 mL·min^−1^. The column temperature was 40 °C.

### 2.7. Determination of Short-Chain Fatty Acids (SCFAs)

An enhanced gas chromatography method was utilized to quantify the SCFA content [25]. The supernatant was obtained by centrifuging 1 mL of fecal fermentation liquid at 12,000 r·min^−1^ for 5 min. Subsequently, 10 µL of internal standard solution (100 mmol·L^−1^ of 2-ethylbutyric acid) and 250 µL of HCl were added to the supernatant. The desired product was then extracted using 1 mL of anhydrous ether. The filtrate was obtained via the addition of anhydrous sodium sulfate to the separated organic phase for dehydration, followed by filtration through a 0.22 µm organic filter membrane. Finally, a 7890A gas chromatograph (Agilent, Santa Clara, CA, USA) was used for the determination of SCFAs. The chromatographic conditions were as follows: HP-INNOWax (Restek, Bellefonte, PA, USA) column with inlet and detector temperatures set at 220 °C and 250 °C, respectively, flow rate of 1.5 mL·min^−1^, heating process from 60 °C to 190 °C for 4 min, and sample (5 µL) to N_2_ ratio of 1:20. The peaks of SCFAs were recorded, and their concentrations were calculated using an external standard method.

### 2.8. Determination of Gut Microbiota

The total genomic DNA of samples was extracted using the cetyltrimethylammonium bromide/sodium dodecyl sulfate method. The specific primers 338F (5′-ACTCCTACGGGAGGCAGCA-3′) and 806R (5′-GGACTACHVGGGTWTCTAAT-3′) were used to amplify the 16S rRNA gene in the V3–V4 region. The sequencing libraries were constructed using NEBNext^®^ Ultra™ IIDNA Library Prep Kit (Cat No. E7645). The chimera sequences were detected using Vsearch (Version 2.15.0) to compare the clean tags with the SILVA database (https://www.arbsilva.de/) (accessed on 14 June 2022). Then, the effective tags were obtained by removing the chimera sequences [26]. Multiple sequence alignment analysis was used to investigate the phylogenetic relationships among amplicon sequence variants (ASVs) and differences in dominant species between various samples or groups.

### 2.9. Determination of Metabolites

The metabolites of different experimental groups were determined using LC–tandem MS (MS/MS) technique. The results were presented in two modes: positive ion (POS) and negative ion (NEG). Statistical software R (R version r-3.4.3), Python (Python 2.7.6), and CentOS (CentOS 6.6) were used for statistical analysis. The Kyoto Encyclopedia of Genes and Genomes (KEGG) database and the Human Metabolome Database were used to annotate various metabolites. The metabolites with variable importance in projection (VIP) >1, *p*-value < 0.05, and fold change ≥2 or ≤0.5 were classified as differential metabolites. The Pheatmap package was used to draw the clustering heatmaps. In R software, two functions, cor () and cor. test (), were used to analyze the correlation and statistical significance between differential metabolites, respectively. The function and metabolic pathways of these metabolites were studied via comparison with the KEGG database.

### 2.10. Statistical Analysis

Three biological replicates were performed for each experiment, and the results were presented as mean ± standard deviation. One-way analysis of variance (ANOVA) and the Duncan multiple test were used to explore the differences and significance between groups in SPSS 26.0 statistical software, respectively. *p* < 0.05 was considered statistically significant.

## 3. Results and Discussion

### 3.1. Molecular Weight Changes of PSA in In Vitro Digestion

Digestive results showed that PSA was unaffected by oral digestion, and its molecular weight did not change significantly (Figure 1). However, after 2 h of gastric digestion, the molecular weight decreased significantly from 260 kDa to about 40 kDa, and a mixed system of PSA and SA was formed. PSA was unstable in an acidic environment, and the strong acidic environment of gastric fluid promoted the hydrolyzation of PSA, which resulted in a significant reduction in its molecular weight. The molecular weight of PSA did not change significantly after 4 h digestion in the small intestine, which indicates the relative stability of PSA in the small intestine fluid. The molecular weight of PSA remained at about 40 kDa after digestion. A mixed system of PSA and SA was formed, and it can reach the colon and be used by the gut microbiota.

### 3.2. Consumption of PSA and SA in In Vitro Fermentation

At 0–12 h, the minimum consumption of PSA was 36.4 mg, which was significantly lower than that of SA (74.4 mg). Within 12–24 h, PSA consumption rapidly increased to 93.6 mg, which was slightly higher than SA consumption (90.0 mg). At 24–36 h, the consumption of PSA increased, but the value was lower than that of SA and reached a maximum of 104.4 mg. Moreover, PSA consumption continuously increased and reached a maximum of 99.6 mg, which was comparable to the SA consumption at 36–48 h (Appendix A). Before entering the experimental period, the gut microbiota was in the exponential growth phase and used various nutrients in the medium. Given that SA is a low-molecular-weight monomer compound, it can be quickly utilized by the gut microbiota to promote bacterial growth. Therefore, the SA group showed a higher consumption in the early stage of fermentation. However, the PSA group was a mixture of SA and PSA with a molecular weight of about 40 kDa. The gut microbiota may lack the enzyme to decompose high-molecular-weight PSA. SA was used for cell growth preferentially, which resulted in the low consumption of the PSA group. With the progress of fermentation, the bacteria grew vigorously, and the amount of enzyme that can secrete PSA for consumption increased. As a result, the consumption of PSA increased.

### 3.3. Effects of SA and PSA on SCFA Content

SCFAs are important metabolites of the gut microbiota, and they play an irreplaceable role in maintaining intestinal homeostasis, immunoregulation, and energy supply. Compared with the blank group, the PSA and SA groups indicated significantly increased content of SCFAs at each fermentation time point (Figure 2). After fermentation for 12 h, the contents of SCFAs in the PSA and SA groups increased rapidly. The contents of total acid in the PSA and SA groups reached 77.59 ± 2.07 and 68.36 ± 0.97 mmol/L, respectively, which were significantly higher than that of the blank group (55.26 ± 0.51 mmol/L). After fermentation for 48 h, the content of total acid in the PSA group increased from 41.27 ± 0.65 mmol/L to 117.67 ± 0.57 mmol/L, and that in the SA group increased from 41.42 ± 0.76 mmol/L to 95.68 ± 0.29 mmol/L, which indicate increases in the content by 48% and 20% compared with the blank group, respectively (Figure 2A).

After 48 h, the acetic acid content of the PSA group increased from 24.19 ± 0.18 mmol/L to 58.15 ± 0.09 mmol/L, and that of the SA group increased from 24.38 ± 0.74 mmol/L to 51.91 ± 0.06 mmol/L; these values were 27% and 13% higher than that of the blank group, respectively (Figure 2B). Acetic acid is the most important substrate for cholesterol synthesis, and it can be ingested and utilized by numerous tissues and participate in the metabolism of muscles, the spleen, the heart, and the brain [27]. The butyric acid content of the PSA group significantly increased from 2.64 ± 0.26 mmol/L to 25.8 ± 0.86 mmol/L, and that of the SA group significantly increased from 2.78 ± 0.24 mmol/L to 18.41 ± 0.13 mmol/L; these values were 239% and 142% higher than that of the blank group, respectively (Figure 2D). Butyric acid has a variety of physiological functions, and it can be absorbed and utilized by colon epithelial cells, regulate gene expression, and maintain the stability of the intestinal environment [28]. Notably, the PSA and SA groups showed different propionic acid contents (Figure 2C). Propionic acid content increased significantly by 23% in the PSA group compared with the blank group. However, no significant increase in propionic acid production of the SA group was observed compared with the blank group. Propionic acid is absorbed by the colon, can reduce cholesterol levels and fat storage, and has anti-cancer and anti-inflammatory properties [28]. PSA and SA can significantly increase the content of SCFAs, which help in maintaining the homeostasis of the intestinal environment and are beneficial to human health. Nevertheless, they showed different results in terms of propionic acid production. Thus, PSA and SA may have different effects on the gut microbiota.

### 3.4. Effects of PSA and SA on Gut Microbiota in In Vitro Fermentation

Non-metric multidimensional scaling (NMDS) analysis results indicated that the clustering trend of the gut microbiota evidently differed among various groups (Figure 3A). Furthermore, analysis of similarities indicated a statistical separation of intestinal microbial community composition among the groups (*p* = 0.005). Similarly, principal coordinate analysis (PCoA) suggested different clustering of gut microbiota in the PSA, SA, and blank groups. The different gut microbiota groups were classified, and samples from the same group were clustered together (Figure 3B). Moreover, permutational multivariate analysis of variance (PERMANOVA) suggested statistical differences between groups (*p* = 0.001). The percentage variations of the first two components were 36.58% and 22.93%, suggesting that PSA and SA had different intestinal microbial community compositions.

At different classification levels, the intestinal microbial community composition was further analyzed. At the phylum level, the gut microbiota of each group mainly consisted of Firmicutes, Bacteroidetes, Proteobacteria, and Fusobacteria (Figure 3C). After 12 h of fermentation, the relative abundance of Bacteroidetes significantly increased, whereas that of Firmicutes significantly decreased in the SA and PSA groups. As fermentation progressed, the levels of Bacteroidetes in the PSA group and Firmicutes in the SA group gradually increased. After 48 h, the PSA group exhibited significantly decreased relative abundances of Proteobacteria and Fusobacteria and increased relative abundances of Bacteroidetes and Firmicutes compared with the blank group. In the SA group, the levels of Proteobacteria, Bacteroidetes, and Fusobacteria significantly decreased, whereas that of Firmicutes increased. The results indicated that Bacteroidetes may be the dominant member of the microbiota that can preferentially utilize SA. In general, PSA and SA enriched Bacteroidetes and Firmicutes, with the latter and former showing a higher relative abundance in the SA and PSA groups, respectively. Furthermore, PSA and SA can significantly reduce Proteobacteria and may have the ability to inhibit the propagation of harmful bacteria [29].

At the genus level, the relative abundances of *Bacteroides*, *Faecalibacterium*, and *Anaerostipes* in the PSA group increased significantly compared with the blank group after 48 h (Figure 3D). *Faecalibacterium* is one of the most crucial bacterial genera in the human gut microbiota; it can produce butyric acid, maintain the balance of the gut microbiota, and protect the colon from intestinal pathogens [30]. In addition, *Anaerostipes* is a common beneficial bacterium in the intestinal tract; it can ferment polysaccharides and produce SCFAs and is an important member of bacteria that produce butyrate in the intestinal tract [31]. The SA group showed significantly increased relative abundances of *Bifidobacterium*, *Megamonas*, and *Lachnoclostridium*. *Bifidobacterium* is an important beneficial member of the gut microbiota and has a series of physiological effects, such as synthesis of vitamins, stimulation of immune function, and improvement of host anti-infection and anti-tumor functions [32]. *Megamonas* can ferment various carbohydrates and produce SCFAs, lactic acid, and other beneficial organic acids, which can maintain the homeostasis of the intestinal environment [33]. Moreover, PSA and SA can inhibit the colonization of potential pathogenic bacteria *Fusobacterium* and *Escherichia-Shigella*. These results suggest that PSA and SA can regulate the community composition of the gut microbiota in different ways, such as by inhibiting the growth of harmful pathogens, promoting the proliferation of different kinds of probiotics, and enabling the gut microbiota to develop toward an orientation beneficial to the human body.

The results of Spearman correlation analysis suggested that total SCFAs, acetic acid, and propionic acid were positively correlated with Bacteroidetes, whereas butyric acid was negatively correlated with *Actinomyces* at the phylum level. At the genus level, butyric acid, propionic acid, and acetic acid were positively correlated with *Anaerostipes*, *Faecalibacterium*, and *Coprococcus* but negatively correlated with *Enterobacter* and *Streptococcus* (Appendix A). In general, changes in microbial communities at the genus level can cause changes in SCFA concentrations in the fermentation broth.

### 3.5. Non-Targeted Metabolomics Analysis of Differential Metabolites

When carbohydrates affect the composition of the intestinal microbial community, the phenomenon will be accompanied by a series of changes in various metabolites. In this study, a non-targeted metabolomics technique was used to investigate the initial point of fermentation and the changes in metabolites in each experimental group after 48 h of fermentation. The results of principal component analysis (PCA) suggested that biological duplicate samples for each group were clustered together, and samples in different experimental groups were separated. Thus, each experimental group exhibited significant differences in metabolites. The addition of PSA and SA changed the metabolic profile. Furthermore, quality control (QC) samples were gathered together, and they showed good data quality and high reliability of the results (Appendix A).

The results of differential metabolite screening showed that 403 metabolites with significant differences were screened in comparison with the blank group (207 in the SA group and 196 in the PSA group). Moreover, 128 differential metabolites were screened in the PSA group in comparison with the SA group. The differential metabolites were analyzed by hierarchical clustering analysis (HCA) to normalize and cluster relative quantitative values. The metabolites of the three experimental groups were significantly enriched at the initial point of fermentation but significantly decreased after 48 h. The concentrations of metabolites in the PSA, SA, and blank groups were significantly different (Appendix A). The results suggest that PSA and SA had significant effects on the metabolism level of related metabolites to varying degrees.

Based on the screening results of differential metabolites, matchstick charts were drawn for the top 20 differential metabolites. The results suggest that in the POS mode, in contrast to the blank group, the PSA group significantly upregulated metabolites, such as 7-(2-aminophenyl) heptanoic acid, ergosterol peroxide, and N-acetyl-L-histidine, and downregulated metabolites, such as N-acetylputrescine, thromboxane B2, and 1,3-dimethyluracil (Figure 4A). In the NEG mode, the significantly upregulated metabolites included xanthurenic acid, tricarballylic acid, and gallic acid. The significantly downregulated metabolites comprised o-toluic acid, phenylpyruvic acid, and cinnamic acid (Figure 4B). Ergosterol peroxide is a steroid derivative that inhibits the inflammatory response of cells and the growth of cancer cells; it has various biological activities, such as anti-tumor, pro-apoptosis, anti-inflammatory, antibacterial, and anti-proliferation effects [34]. Gallic acid has anti-inflammatory, anti-mutation, anti-oxidation, anti-free-radical, antiviral, and other biological activities and is beneficial to human health [35]. In addition, putrescine is a kind of metabolite that is harmful to the human body, can damage the intestinal epithelial cells and intestinal wall, and can increase intestinal permeability, resulting in a series of physiological or pathological changes in the body. PSA can upregulate beneficial metabolites, such as ergosterol peroxide and gallic acid, and downregulate the harmful substance N-acetylputrescine, which may have a positive effect on human health.

The SA group showed significantly upregulated levels of 5-methyl-2′-deoxycytidine, 4-pyridoxic acid, and normorphine and downregulated levels of palmitoylethanolamide, 2-phenylethlamine, and tryptophanol compared with the blank group in the POS mode (Appendix A). The significantly upregulated metabolites were amentoflavone, capryloylglycine, and lipoic acid, and the downregulated ones included thymidine 5′-monophosphate, homovanillic acid, and cinnamic acid in the NEG mode (Appendix A). 4-Pyridoxic acid has a strong antioxidant capacity, can promote the synthesis of nucleic acid, prevent the aging of tissues and organs, and promote tryptophan conversion to niacin [36]. Lipoic acid can be used as a coenzyme participating in acyl transfer in the metabolism of substances in the body; it has significant electron affinity and the capability to react with free radicals. Lipoic acid, with antioxidant properties, can also eliminate free radicals that lead to accelerated aging and disease and thus has extremely high health function and medical value [37]. SA significantly upregulated beneficial metabolites, such as 4-pyridoxic acid and lipoic acid, which indicates that it can upregulate metabolites with antioxidant function, maintain body balance, and delay aging.

Furthermore, compared with the SA group, the PSA group exhibited a significant upregulation of anandamide (AEA), methionine, and 16,16-dimethyl prostaglandin A1 and downregulation of flavanone, theobromine, and N-acetylglucosamine 1-phosphate in the POS mode (Appendix A). In the NEG mode, PSA can significantly upregulate docosahexaenoic acid, docosatrienoic acid, and syringic acid and downregulate phenylpyruvic acid, palmitoleic acid, and deoxycholic acid (Appendix A). AEA is an endogenous cannabinoid that can regulate the functions of central and peripheral nervous and immune systems to maintain the physiological balance of the body [38]. Methionine is an essential amino acid that cannot be synthesized by the human body, but it has various physiological functions, such as liver protection, anti-depression, blood pressure reduction, and metabolism promotion [39]. Intestinal microbial metabolites have a wide range of effects on human health [40]. The results showed that PSA and SA had different effects on intestinal microbial metabolites. Moreover, PSA and SA can positively regulate intestinal microbial metabolites and may positively contribute to maintaining human health.

The results of KEGG enrichment analysis suggested that in contrast to those in the blank group, the main metabolic pathways in the PSA group were amino acid (histidine and alanine metabolism) and purine metabolisms (Figure 5A). Histidine can help in the formation of healthy tissues in the body; it covers the myelin sheath of nerve cells and ensures the smooth transmission of information from the brain to all parts of the body. Although histidine can be synthesized in the adult body, its deficiency easily occurs, which affects human health. Moreover, histidine produces histamine and can promote iron absorption, which has physiological functions such as the prevention of anemia. PSA mainly affected the histidine synthesis pathway in the histidine metabolism pathway. Specifically, PSA caused the upregulation of carnosine by causing the downregulation of anserine and ultimately affected the synthesis of L-histidine, which resulted in the upregulation of histidine (Appendix A). Amino acids are involved in numerous important physiological activities in the body and make an important contribution to maintaining human health. PSA can promote the metabolism of amino acids and has a positive effect on promoting human health.

The main metabolic pathways enriched in the SA group were nicotinate and nicotinamide metabolism, pyrimidine metabolism, and purine metabolism (Figure 5B). Nicotinate and nicotinamide are some of the essential micronutrients in the body, and they have multiple physiological functions, such as antioxidant and anti-inflammatory effects and nervous health maintenance. In the metabolic pathway of nicotinate and nicotinamide, SA can upregulate trigonelline and succinate and downregulate nicotinamide and nicotinurate, which affect the metabolism of niacinamide and niacinamide (Appendix A). SA can promote the metabolism of niacin and niacinamide, which may enhance the antioxidant-related effect and contribute to the promotion of human health.

Furthermore, compared with the SA group, the main metabolic pathways in the PSA group included tyrosine metabolism, biosynthesis of unsaturated fatty acids, and glutathione metabolism (Appendix A). Tyrosine is a conditionally essential amino acid for the human body, and it can regulate mood, stimulate the nervous system, and help speed up metabolism. PSA affected the metabolism of tyrosine by upregulating L-DOPA (Levodopa) and fumarate and downregulating L-tyrosine and acetoacetate (Appendix A). The enrichment results of the metabolic pathway showed that the addition of SA and PSA changed the metabolic pathway of the gut microbiota.

In conclusion, the 16s rDNA and non-targeted metabolome results strongly support our hypothesis that PSA and SA affect the gut microbiota and metabolites in different ways and contribute positively to the maintenance of human health.

### 3.6. Correlation Analysis between Differential Metabolites and Microbiota

Correlation analysis of non-targeted metabolome and microbiome data was performed to investigate metabolite changes that might have been caused by changes in microbial community structure. Correlation analysis was conducted between the top 10 differential microbiota at the genus level and the top 20 differential metabolites based on the Pearson correlation coefficient, and a heat map was drawn to determine the degree of correlation between microbiota diversity and metabolites. The results between the PSA and blank groups showed that *Enterobacter* and *Fusobacterium* were positively correlated with tryptophanol, 6-dimethyl-9H-purin-6-amine, and creatine phosphate and showed a significant negative correlation with 5-methyl-2′-deoxycytidine, 5,7-dihydroxy-3-(4-hydroxyphenyl)-4H-chromen-4-one, and L-leucyl-L-alanine in the POS mode (Figure 6A). *Enterobacter* and *Fusobacterium* were positively correlated with thymidine 5′-monophosphate, homovanillic acid, and lipoic acid and negatively correlated with syringic acid, tricarballylic acid, and N-acetyl-L-leucine in the NEG mode (Figure 6B). Nevertheless, *Escherichia-Shigella*, *Veillonella*, *Allisonella*, *flavonifractor*, *Lachnoclostridium*, *Sutterella*, and *Odoribacter* showed opposite correlations with the above metabolites compared with *Enterobacter* and *Fusobacterium*.

The results between the SA and blank groups suggest that *Fusobacterium*, *Bacteroides*, *Enterobacter*, and *Parabacteroides* were positively correlated with oleanolic acid, tryptophanol, and acetylcarnitine and significantly negatively correlated with 4-pyridoxic acid, L-leucyl-L-alanine, and deoxycytidine in the POS mode (Figure 6C). In addition, in NEG mode, these bacteria were positively correlated with 2-hydroxyvaleric acid, pregnenolone, and deoxyguanosine and negatively correlated with LPE 18:2, 3-methyl-5-oxo-5-(4-toluidino) pentanoic acid, and N-acetylglycine (Figure 6D). By contrast, *Megamonas*, *Subdoligranulum*, *Lachnoclostridium*, and *Flavonifractor* had a completely opposite correlation with the above-mentioned metabolites.

Moreover, the results between PSA and SA groups showed that *Veillonella*, *Allisonella*, *Flavonifracto*, *Parabacteroides*, and *Phascolarctobacterium* had a significantly negative correlation with pantothenic acid, AEA, and linoleoyl ethanolamide and a negative correlation with 1,4-dihydroxyheptadec-16-en-2-ylacetate, 1,3-dimethyluric acid, and methyl alpha-D-glucopyranoside in the POS mode (Appendix A). In the NEG mode, these bacteria were positively correlated with syringic acid, 2-hydroxy-2-methylbutanedioic acid, and uric acid and negatively correlated with corchorifatty acid F, palmitoleic acid, and cyclohexaneacetic acid (Appendix A). However, *Megamonas*, *Succinatimonas*, and *Dorea* showed opposite correlations for the same metabolites compared with the above bacteria.

Altogether, these results reveal a potentially important relationship between significantly altered metabolites and microbial composition after the addition of PSA and SA and may lead to new research hypotheses regarding the existence of specific microchemical associations.

## 4. Conclusions

In this study, BGR was used for the first time to investigate the digestion of PSA and explore the effects of the addition of PSA and SA on gut microbiota composition and metabolites in healthy people. The results suggest that PSA can retain the active components after digestion. Moreover, the addition of PSA and SA can regulate the gut microbiota composition; it increased the relative abundances of Firmicutes and Bacteroides and decreased those of Proteobacteria and Fusobacteria. Meanwhile, the content of SCFAs increased significantly. Furthermore, PSA and SA affected the gut microbiota community composition and metabolites in different ways, which resulted in positive effects on human health. This study indicated that PSA and SA have a potential application prospect in the development of health functional foods. In the future, in vivo animal studies will be conducted to further evaluate whether dietary supplementation with PSA and SA is able to change the gut microbiota composition and metabolites.

## Figures and Tables

**Figure 1 foods-13-00481-f001:**
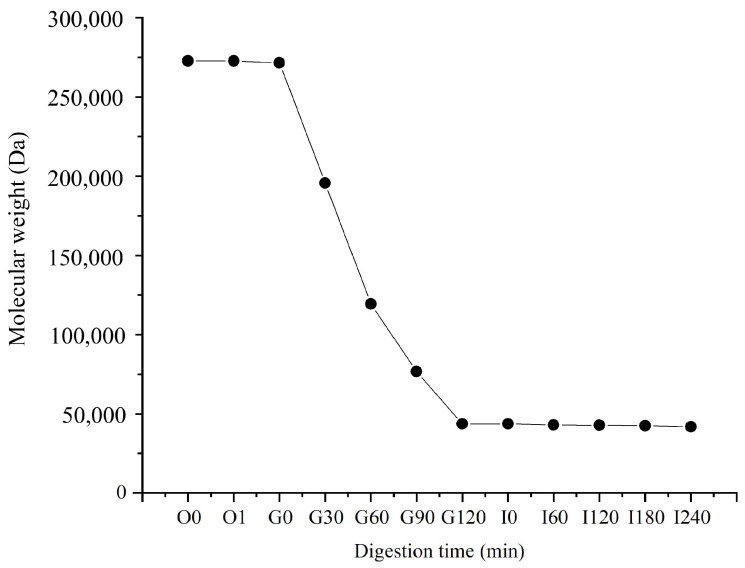
Changes in molecular weight of PSA during digestion. O: oral digestion, G: gastric digestion; I: intestinal digestion.

**Figure 2 foods-13-00481-f002:**
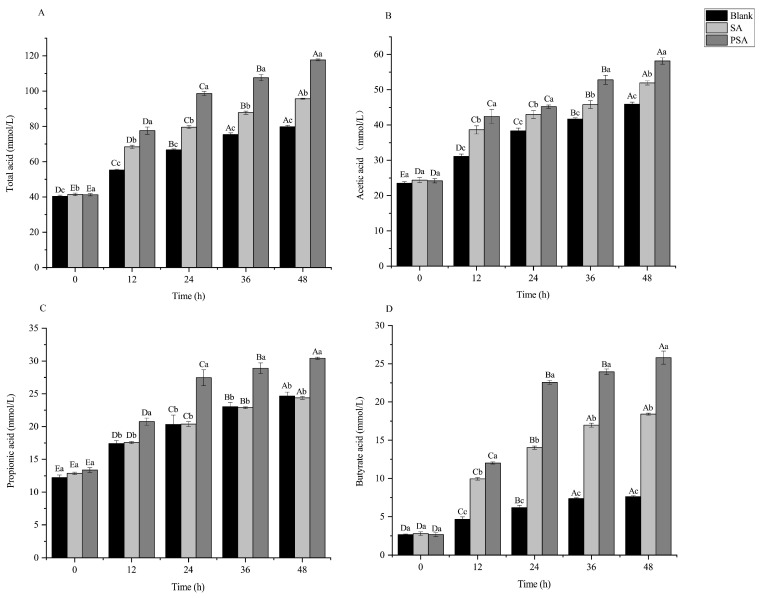
Contents of SCFAs in the PSA, SA, and blank groups under different fermentation times. (**A**) Acetic acid; (**B**) propionic acid; (**C**) butyric acid; (**D**) total acid. Values are means ± SEM. Different capital letters (A, B, C, D, and E) indicate that the same experimental group has significant differences (*p* < 0.05) at different fermentation time points. Different lowercase letters (a, b, and c) indicate that different experimental groups have significant differences (*p* < 0.05) at the same fermentation time point.

**Figure 3 foods-13-00481-f003:**
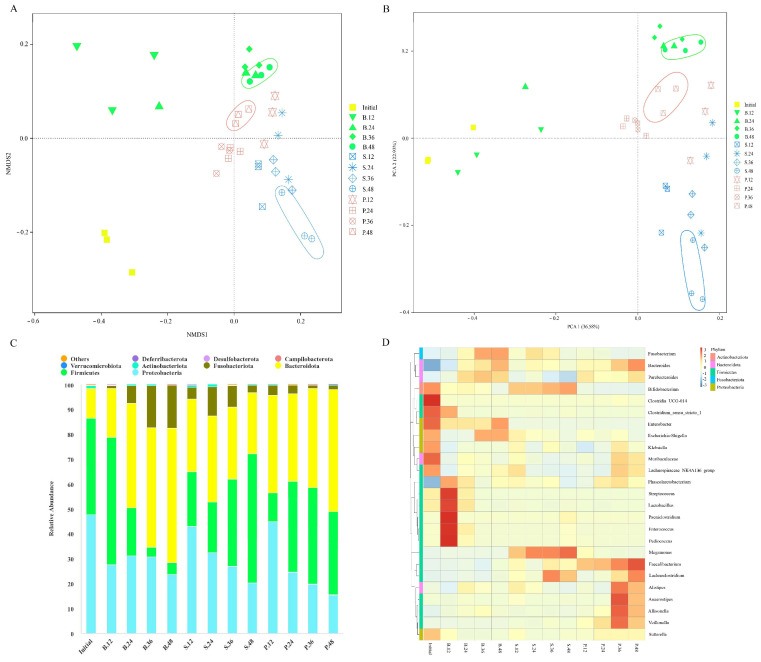
Changes in intestinal microbial community composition. (**A**) NMDS analysis, (**B**) PCoA, (**C**) histogram of abundance distribution at the phylum level, (**D**) cluster heat map of species distribution at the genus level. Initial, B, S, and P represent the initial point of fermentation and blank, SA, and PSA groups, respectively. The numbers 12, 24, 36, and 48 indicate 12, 24, 36, and 48 h of fermentation, respectively.

**Figure 4 foods-13-00481-f004:**
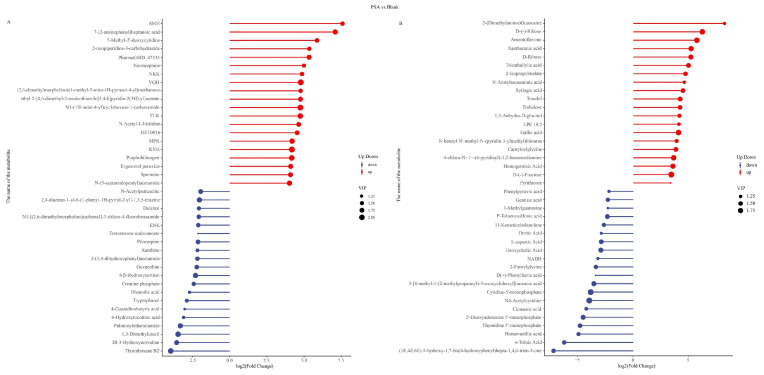
Matchstick charts of the top 20 differential metabolites between the PSA and blank groups. (**A**) POS mode. (**B**) NEG mode. Red and blue colors indicate the upregulated and downregulated metabolites, respectively, and the sizes of the circles refer to different VIP values.

**Figure 5 foods-13-00481-f005:**
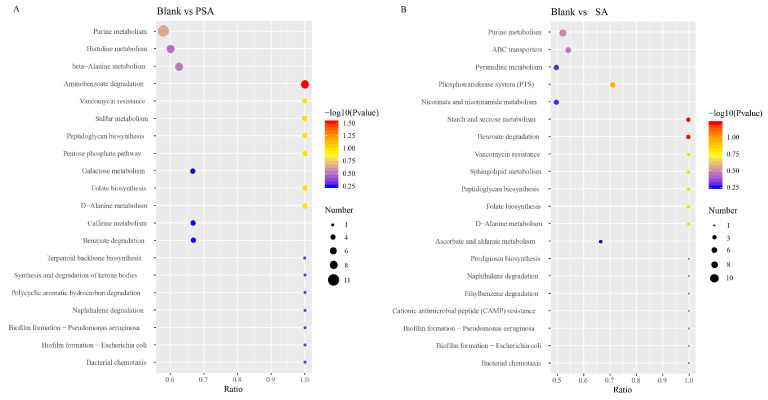
Metabolic pathways associated with differential metabolite changes between different experimental groups. (**A**) Blank group vs. PSA. (**B**) Blank group vs. SA. The x-coordinate is x/y (the number of differential metabolites in the corresponding metabolic pathway/the total number of identified metabolites in the pathway). The color of the dots indicates the *p*-value of the hypergeometric test, and their size denotes the number of differential metabolites in the corresponding pathway.

**Figure 6 foods-13-00481-f006:**
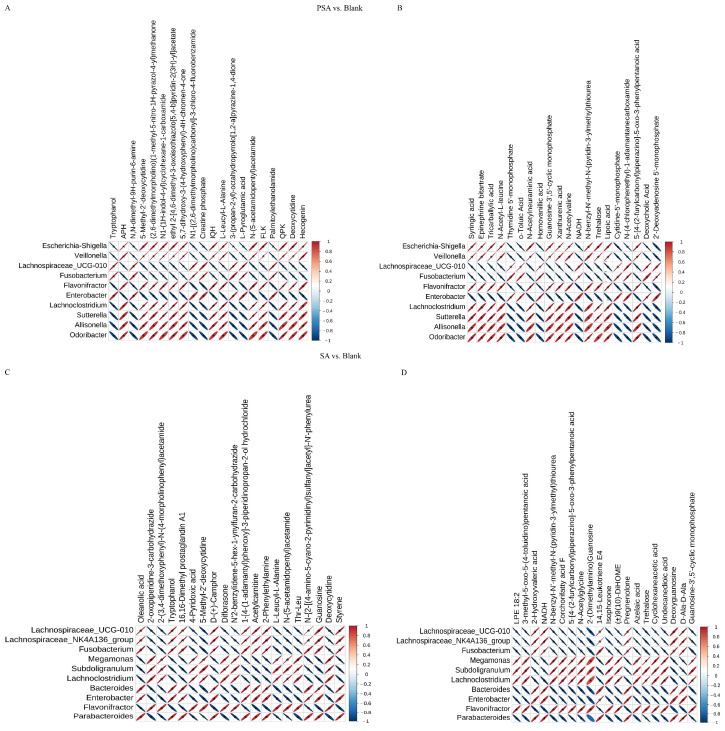
Correlation heat maps of different bacterial genera and metabolites between groups at the genus level. (**A**) POS mode of PSA vs. blank. (**B**) NEG mode of PSA vs. blank. (**C**) POS mode of SA vs. blank. (**D**) NEG mode of SA vs. blank. The abscissa represents the differential metabolites, and the ordinate indicates the differential bacteria. The legends reveal the correlation coefficient, with red color indicating a positive correlation and blue denoting a negative correlation. The asterisk mark means statistical difference; *p* < 0.05. The size of the area of the ellipse represents the size of *p*, and the larger the area, the smaller the *p* value.

## Data Availability

The data are not publicly available due to the privacy limitations of the relevant research. The data presented in this study are available on request from the corresponding author.

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
