# Peer review of "Effects of In Vitro Fermentation of Polysialic Acid and Sialic Acid on Gut Microbial Community Composition and Metabolites in Healthy Humans"

_foods, 2024, doi:10.3390/foods13030481_

Round 1
Reviewer 1 Report
Comments and Suggestions for Authors
The study is interesting and relevant for the field. Some issues in the description of specific items were detected:
Lines 75 and 76, your affirmations are part of the hypothesis rather than predictions.Usually, prediction is used more often for results of a prospective model.
Lines 105-110:"Then, 95% ethanol at three times the volume" it is clearer to refer the ethanol addition as a ratio of 3:1.
Line 108: "the precipitation was collected", the correct word is "the precipitate"
Line 112: It seems that a subheading is missing.
Line 117: how the purity of the SA was confirmed?
Line 127: Better hydrochloric acid instead of muriatic acid.
Line 132: What means "supercarbonate" in this context? It is not a standard name for a chemical compound.
Lines 152-154: The description of media for the experiment is not clear enough.
Lines 243-244: "Before entering the experimental period, gut microbiota was in the hungry stage and
used various nutrients in the medium". It seems to refer to the exponential growth phase, which is a standard way to name this metabolic state.
Section 3.2: was the total biomass quantified? The growth dynamics may reflect if the difference between SA and PSA uptakes affect the growth in each case.
Consider to plot the data of Table 1 as Concentration vs time for each experiment with error bars. It would make easier to read and analyze the data.
Supplementary material was not cited.
Comments on the Quality of English LanguageIt is recommended to double check the redaction style. Although the results and discussion section was clearer than the previous ones.
Author Response
Response to Comments from reviewers
Reviewer #1:
The study is interesting and relevant for the field. Some issues in the description of specific items were detected.
Response: Thank you for your affirmation of our study. Thanks very much for your valuable comments, which can make our manuscript more complete.
- Lines 75 and 76, your affirmations are part of the hypothesis rather than predictions. Usually, prediction is used more often for results of a prospective model.
Response: Thank you for raising this concern. We have corrected “hypothesized” instead of “predicted” (please see line 76).
Lines 105-110:"Then, 95% ethanol at three times the volume" it is clearer to refer the ethanol addition as a ratio of 3:1.
Response: We thank the reviewer for pointing this out to us. We have corrected “95% ethanol was mixed with the supernatant at a volume ratio of 3:1” instead of “95% ethanol at three times the volume was added into the supernatant” (please see line 107).
- Line 108: "the precipitation was collected", the correct word is "the precipitate".
Response: We appreciate you point out this. We have corrected “precipitate” instead of “precipitation” (please see line 108).
- Line 112: It seems that a subheading is missing.
Response: We appreciate you point out this. Section 2.2 is about the preparation and extraction of PSA and SA, including the preparation and extraction of PSA and the preparation and extraction of SA, respectively. Therefore, line 112 is included in section 2.2, without missing a subheading.
- Line 117: how the purity of the SA was confirmed?
Response: Thank you for raising this concern. After polysialic acid was hydrolyzed by formic acid, the hydrolysate was extracted and purified, and the purity of sialic acid was determined by high performance liquid chromatography (HPLC).
- Line 127: Better hydrochloric acid instead of muriatic acid.
Response: Thank you for raising this concern. We have corrected “hydrochloric acid” instead of “muriatic acid” (please see line 129).
- Line 132: What means "supercarbonate" in this context? It is not a standard name for a chemical compound.
Response: Thanks for your valuable comments. We have corrected “bicarbonate” instead of “supercarbonate” (please see line 133).
- Lines 152-154: The description of media for the experiment is not clear enough.
Response: Thank you for your professional advice, and we apologize for the ambiguous narrative. We have corrected “For the supplemental medium, the composition of the blank group was exactly the same as that in the basic medium. While, in the PSA (SA) group, the amylum in the supplemental medium was replaced by PSA (SA), and the remaining components were the same as those in the blank group” instead of “The composition of supplemental medium in the blank group was exactly the same as that in the basic medium, whereas the carbon sources (amylum) in the supplemental medium of SA and PSA groups were replaced by SA and PSA, respectively.” (please see lines 154-157).
- Lines 243-244: "Before entering the experimental period, gut microbiota was in the hungry stage and used various nutrients in the medium". It seems to refer to the exponential growth phase, which is a standard way to name this metabolic state.
Response: Thanks for your valuable comments. We have corrected “the exponential growth phase” instead of “hungry stage” (please see line 248).
- Section 3.2: was the total biomass quantified? The growth dynamics may reflect if the difference between SA and PSA uptakes affect the growth in each case.
Response: Thank you for your professional advice. During the experiment, OD600 of fermentation broth at 0,12,24,36 and 48h were measured to observe the growth of gut microbiota. The results showed that the growth of gut microbiota in blank group, PSA group and SA group was good. However, it is not possible to determine which gut microbiota make use of PSA and SA based on changes in OD. Therefore, 16s sequencing was performed in order to more intuitively demonstrate the relationship between substrate consumption and gut microbial growth.
- Consider to plot the data of Table 1 as Concentration vs time for each experiment with error bars. It would make easier to read and analyze the data.
Response: Thank you for your professional advice. We removed the original “table 1” and added a new “Figure 2” to the section 3.3 (please see lines 289-295).
- Supplementary material was not cited.
Response: We apologize for this situation. At the time of uploading the manuscript, we have uploaded supplementary material to the website. We have re-added supplementary material to the manuscript for your review.
Reviewer 2 Report
Comments and Suggestions for Authors
In this paper, the authors analyzed the influence of polysialic acid and sialic acid on the gut microbial community composition and metabolites in healthy human, using a bionic gastrointestinal reactor, demonsatrting that PSA and SA significantly changed the gut microbiota and metabolites in different degrees.
The paper is quite well written and the results presented are interesting.
Nevertheless, some clarifications and corrections are needed.
In Materials and methods, section 2.2 and 2.3: the centrifugation parameter " r x min -1" should be corrected and replaced by rcf or g
section 2.3: the simulated saliva composition should be described
section 2.9, line 212: Cor () and Cor.mtest (), please explain the meaning of empty parenthesis
In the Results, Fig 2 and 3 have poor quality and they are not easily readable
section 3.5, please explain the meaning of NEG mode and POS mode
Fig.4, please correct the B panel description: it is blank vs SA (instead of blank vs PSA)
Fig.5: please, explain what means the different thickness of the lines red and blue
Conclusions: in this section the specification of further studies is needed to assess if the diet supplementation of PSA and SA could be able to change the gut microbiota composition and metabolites
Comments on the Quality of English LanguageMinor editing of English language is required
Author Response
Response to Comments from reviewers
Reviewer 2:
In this paper, the authors analyzed the influence of polysialic acid and sialic acid on the gut microbial community composition and metabolites in healthy human, using a bionic gastrointestinal reactor, demonsatrting that PSA and SA significantly changed the gut microbiota and metabolites in different degrees.
The paper is quite well written and the results presented are interesting.
Response: Thank you for your comments. We are honored to receive your recognition of our research and writing.
Nevertheless, some clarifications and corrections are needed.
- In Materials and methods, section 2.2 and 2.3: the centrifugation parameter " r x min -1" should be corrected and replaced by rcf or g.
Response: Thanks for your valuable comments. We have converted “10000 r·min-1” and “12000r/min” to “9391g” and “13363g” to the section 2.2 (please see lines 107,108 and 110).
- section 2.3: the simulated saliva composition should be described.
Response: Thanks very much for your professional comment and helpful suggestion. We have added the content “The simulated saliva was composed of 6.2 g·L-1 NaCl, 2.2 g·L-1 KCl, 1.5 g·L-1 CaCl2·2H2O, and 5.5 mg·L-1 α-amylase.” to the “section 2.3” (please see lines 122-123).
- section 2.9, line 212: Cor () and Cor.mtest (), please explain the meaning of empty parenthesis
Response: Thank you for your very careful and responsible comments. We have added the content “In R software, two functions Cor () and cor. test ()” (please see line 215). In R software, the cor () and cor. test () functions were used to calculate the correlation and significance between two variables, respectively. In the case of cor. test (), the empty parentheses () indicated that the function took two arguments and wanted to calculate the two data sets for correlation. In the command of cor.test (x, y, method = " Pearson "), x and y were the two variables to be tested, and the method parameter specified the Pearson correlation coefficient as the specific correlation coefficient type. The empty parentheses in cor () represent the same meaning as above.
- In the Results, Fig 2 and 3 have poor quality and they are not easily readable.
Response: Thanks very much for your constructive comment and helpful suggestion. We have improved the clarity of the original figures 2 and 3 (new figures 3 and 4) so that they can be read clearly.
- section 3.5, please explain the meaning of NEG mode and POS mode.
Response: Thanks for your valuable comments. pos and neg represent two modes of positive and negative ions respectively during data acquisition. Positive and negative ion modes are two scanning modes of mass spectrometry. Positive and negative ion modes are two scanning modes of mass spectrometry. After the sample is ionized by ESI source, positively and negatively charged ions will appear at the same time. According to the differences in physical and chemical properties of substances, some metabolites will be positively charged and some will be negatively charged. In general, in order to obtain comprehensive metabolome information, ions in both states of the LC-MS non-targeted metabolome study are scanned. We have added " The results were presented in two modes: positive ion (POS) and negative ion (NEG)" to the section 2.9 in line 209.
- Fig.4, please correct the B panel description: it is blank vs SA (instead of blank vs PSA).
Response: Thanks very much for your professional comment and helpful suggestion. We have corrected “Blank vs SA” instead of “Blank vs PSA” in the Fig. 5.
- Fig.5: please, explain what means the different thickness of the lines red and blue.
Response: Thanks very much for your professional comment and helpful suggestion. In Figure 6, the different thickness of the lines red and blue indicated the size of the P-value, and the thicker the line, the smaller the P-value. We have added “The size of the area of the ellipse represented the size of P value, and the larger the area, the smaller the P value.” to the caption of Figure 6 (please see lines 505-506).
- Conclusions: in this section the specification of further studies is needed to assess if the diet supplementation of PSA and SA could be able to change the gut microbiota composition and metabolites.
Response: Thank you for raising this concern. In the future, in vivo animal studies will be conducted to further evaluate whether the diet supplementation of PSA and SA could be able to change the gut microbiota composition and metabolites. We have added “In the future, in vivo animal studies will be conducted to further evaluate whether the diet supplementation of PSA and SA could be able to change the gut microbiota composition and metabolites.” to the “Conclusions” section (please see lines 521-523).